# The Potential of TRPA1 as a Therapeutic Target in Cancer—A Study Using Bioinformatic Tools

**DOI:** 10.3390/ph17121657

**Published:** 2024-12-09

**Authors:** Dana Cucu

**Affiliations:** 1Department of Anatomy, Biophysics and Physiology, Faculty of Biology, University of Bucharest, Spl. Independentei 91-95, 050095 Bucharest, Romania; dana.cucu@bio.unibuc.ro; 2Research Institute of the University of Bucharest (ICUB), University of Bucharest, 90-92 Sos. Panduri, 050663 Bucharest, Romania

**Keywords:** TRPA1, 4-HNE, PDAC, pancreatic adenocarcinoma

## Abstract

**Background:** The expression of the transient receptor potential 1 (TRPA1) gene is increased in many solid tumours, and its function relates to inflammation, oxidative stress or the presence of toxic substances. However, little is known about the correlation of clinical parameters with patients’ cancer stages, metastases and the degree of tumour infiltration by immune cells. **Methods:** We performed a bioinformatic analysis, using databases and public resources to investigate TRPA1 for many available samples. We used samples from the TCGA project and quantified the mRNA expression and survival analysis using TIMER, TIMER.2 and GEPIA. To analyse hypermethylation, a more extensive database was available from the UALCAN website. **Results:** We show that the TRPA1 gene is hypermethylated in many cancers. The high expression of TRPA1 is correlated with a better prognosis for several cancer types and correlates with cancer stage and metastasis, while in others the TRPA1 is pro-oncogenic. We also report the effect of TRPA1 expression in immune infiltrating cells. Moreover, the expression is linked to genes essential for inflammation, oxidative stress and cellular motility processes. **Conclusions:** Our study brings new insights into the regulation of TRPA1 expression in different tumours based on analysis provided by public databases, opening the possibility to further investigate the protein as a putative target for cancer.

## 1. Introduction

Transient receptor potential 1 (TRPA1) is the only member of the TRPA family from the TRP superfamily of membrane proteins. Its function was first described in 2003 by Story et al., who identified it as a receptor for noxious cold temperatures [1]. The tremendous number of studies that came later have presented the TRPA1 channel as a sensor for inflammation [2], oxidative stress [3] or toxic substances [4]. Because these cellular processes are also pathways in the onset or development of tumours, studying TRPA1 in tumorigenesis was just an intuitive step.

Of all TRP members, TRPA1 is the most abundant redox channel in many solid tumours. From studies on cell cultures, we know that TRPA1 expression is elevated in pancreatic adenocarcinoma [5], nasopharyngeal carcinoma [6], osteosarcoma [7], metastatic colorectal carcinoma [8] and glioblastoma [9]. These studies also commonly analyse the functionality of TRPA1, its activation or inhibition and how it modulates the entry of Ca^2+^ into cells.

The structure of TRPA1 enables activation by electrophilic compounds such as mustard, garlic and cinnamaldehyde and non-electrophilic substances like menthol, carvacrol and clotrimazole. Its complex reactive cysteine residue profile allows TRPA1 to sense H_2_O_2_-induced mitochondrial damage and apoptosis and may have a therapeutic role in TMZ-resistant glioblastoma [10]. Additionally, 4-HNE, a product from lipid peroxidation, activates TRPA1 in pancreatic adenocarcinoma by redox modification of the free thiol groups [11].

Although the studies are substantial and foresee TRPA1 as a putative therapeutic target in cancer, the results do not match the analysis of patient tumours. In a previous study on pancreatic adenocarcinoma patients, we presented the analysis of many TRP channels, including TRPA1 [12]. So far, we do not know whether TRPA1 expression is different in tumours than in non-tumoral samples for other cancer types, and we do not know whether the expression is related to the clinical parameters of patients. In this analysis, we intend to offer some clues regarding the expression of TRPA1 in patients related to overall survival, cancer stage and metastasis, immune infiltration and correlation with genes essential in inflammation and oxidative stress.

## 2. Results

### 2.1. TRPA1 Gene Expression and Prognostic Value in Pan-Cancer Patients

We investigated the mRNA levels of TRPA1 using TIMER.2 analysis, which investigates tumour and normal samples from the TCGA project. Figure 1 illustrates the differential expression of TRPA1 over 31 malignancies with 8112 tumoral samples and 1520 normal samples.

From these data, we deduced that TRPA1 mRNA expression was significantly upregulated (Wilcoxon test, *p* < 0.05) in tumours from invasive breast carcinoma (BRCA), cholangiocarcinoma (CHOL), head and neck squamous cell carcinoma (HNSC), kidney renal clear cell carcinoma (KIRC), kidney renal papillary cell carcinoma (KIRP), lung adenocarcinoma (LUAD), lung squamous cell carcinoma (LUSC), prostate adenocarcinoma and (PRAD) and thyroid carcinoma (TCHA).

Cancers in which TRPA1 is significantly underexpressed, and therefore may act as a tumor suppressor, are bladder urothelial carcinoma (BLCA), colon adenocarcinoma (COAD), kidney chromophobe (KICH), glioblastoma multiforme (GBM) and rectum adenocarcinoma (READ).

To further investigate the link between TRPA1 expression and cancer patients’ overall survival, we performed the survival analysis using GEPIA 2. This analysis revealed that TRPA1 overexpression is related to overall survival only in renal clear cell carcinoma (KIRC) (Figure 2A). Interestingly, high expression is significantly associated with a favourable prognosis in this cancer type.

The correlation of TRPA1 expression with disease-free survival (DFS) returns a significant value (Mantel–Cox test, *p* < 0.05) for KIRC and sarcoma (SARC) (Figure 2B). Again, the high expression indicates a good prognosis in KIRC. Yet, it significantly correlates with the unfavourable prognosis in sarcoma (SARC).

Because TRPA1 gene expression does not influence the patients’ prognosis, we decided to analyse the correlation of TRPA1 with the cancer stages (S2–S4) using the TIMER.2 resources.

### 2.2. TRPA1 Expression Correlates with Cancer Stages and Metastasis

Figure 3 illustrates the heat map that reports the *p*-values and the direction of the association between the differential expression of TRPA1 and the cancer stages. Then, patients’ risk score was calculated based on gene expression and the corresponding Cox regression coefficient, as indicated in red (increased risk) and blue (decreased risk). Also, we provide the Kaplan–Meier survival curves categorised into low-expression and high-expression levels of TRPA1. From these data, differential TRPA1 expression strongly correlated with high cancer stages (3 and 4) in BRCA, COAD, ESCA, KICH, KIRP, KICH, LIHC, LUAD and STAD, but also with skin cutaneous melanoma (SKCM), which was not significant in our previous analysis.

A moderate correlation was noticed for BRCA-Her2, lung adenocarcinoma (LUSC), stomach adenocarcinoma (STAD), thyroid cancer (THCA) and uveal melanoma (UVM). Although these results highlight the expression of TRPA1 as important in cancer progression from stage 1 to stage 3, the results are mirrored by significant changes in clinical parameters such as overall survival only in BRCA-Her2, KICH and READ cancers, where TIMER.2 returns a z score indicating a good prognosis (bold red values, left side of Figure 3B). An increased risk (*p* < 0.05) and, consequently, a poor prognosis is returned for KIRC, LUAD, PRAD and THYM, illustrated in bold values on the left side of Figure 3B.

Next, we examine the correlation of TRPA1 expression and metastasis from the tnmplot dataset, which includes, after curation, 11,010 samples, with 730 normal, 9886 cancerous and 394 metastatic specimens from The Cancer Genome Atlas (TCGA), the Therapeutically Applicable Research to Generate Effective Treatments (TARGET), the Genotype-Tissue Expression (GTEx) and the Gene Expression Omnibus of National Center for Biotechnology Information (NCBI-GEO) databases [13]. We compared the expression of TRPA1 from gene chip analysis in normal, tumoral and metastatic tissues. The results are illustrated in Figure 4A and indicate significantly higher expression in intestine, lung and skin cancers in tumours and metastatic samples than in normal non-tumoral tissues. Most other cancer tissues revealed substantially lower expression in tumoral samples than the normal, non-tumoral ones. Moreover, although the samples had a higher TRPA1 expression in breast, prostate and ovarian cancer tumours, the metastatic tissues recovered almost entirely under the initial values from the non-tumoral tissues. Figure 4B illustrates the median fold changes (FCs) for TRPA1 expression in tumoral vs. normal and metastatic vs. tumoral tissues and the *p*-values resulting from performing two-tailed Dunn’s tests between these samples. TRPA1 was most consistently upregulated with higher FC values (from 2.23 to 3.34) in metastatic samples from colon, intestine, lung and skin samples.

### 2.3. TRPA1 Expression Is Modified in Tumor-Infiltrating Immune Cells

Increased evidence supports the hypothesis that the infiltration of various immune cells increases metastasis ability. Using TIMER, we determined the Spearman correlation between TRPA1 expression levels and tumour purity. The scatterplots displayed in Figure 5 show the purity-corrected partial Spearman’s rho value and statistical significance. We examined the relationship between TRPA1 expression and tumour samples, which indicated a high expression in our previous analyses. Genes highly expressed in the microenvironment are expected to have negative associations with tumour purity, while the opposite is expected for genes highly expressed in the tumour cells.

Negative and significant correlations between TRPA1 expression and tumour purity were found for COAD (*p* = 8.04 × 10^−3^), LUAD (*p* = 2.28 × 10^−7^) and PRAD (*p* = 0.37 × 10^−7^). In contrast, significant positive correlations were found for KICH (*p* = 1.50 × 10^−2^) and KIRP (*p* = 7.92 × 10^−3^).

We extracted Kaplan–Meier plots for immune infiltrates and TRPA1 from TIMER to visualise the survival differences (Figure 6). We applied the correction for tumour purity, and expression levels were divided into low and high groups. TIMER returned a good prognosis for a high level of TRPA1 expression in the following immune cells: B cells for BRCA-Her2, LUAD, SKCM and THYM cancers; T CD8+ cells from ACC and SKCM cancer; T CD4+ cells for CESC; macrophages from THYM cancer; neutrophils from CHOL, SKCM and UMV cancer; and dendritic cells in SKCM and THYM. An unfavourable prognosis was found for high TRPA1 expression in B cells from KIRC, T CD8+ cells from BLCA, KIRC, UMV macrophages from STAD and dendritic cells from GBM.

### 2.4. Gene Promoter Methylation

So far, our analysis has been restricted to gene expression. Still, an important aspect of TRPA1 protein is its functionality, revealed by activation or inhibition by specific agonists and antagonists. Moreover, the function of this receptor can be modulated by posttranslational modification or TRPA1 promoter methylation.

Based on the UALCAN online tool, the promoter region of TRPA1 was found to be significantly (Student’s *T*-test, *p* < 0.05) hypermethylated compared to non-tumoral samples in samples from BLCA, BRCA, CHOL, COAD, HNSC, KIRP, LUAD, LUSC, PRAD, READ and TCHA. Figure 7 illustrates the methylation of the TRPA1 gene promoter in normal and tumour samples. The table on the right side of the figure indicates the *p* coefficient extracted from Student’s *t*-test.

### 2.5. TRPA1 Expression Correlates with Genes Important in Inflammation, Oxidative Stress and Cellular Migration

TRPA1 mainly participates in the inflammatory process and oxidative stress when inflammatory cytokines modulate their expression [14] or directly through electrophilic stimuli. TRPA1 also participates in the migration and proliferation of some cancer cells [5]. To visualise the correlation of TRPA1 expression with different genes involved in these processes, we employed TIMER.2 resources. Based on previous studies [3,15,16], we chose the following genes for analysis: the transcription factor Paired Box 6 (PAX6), Ankyrin 1 (ANK1), Calmodulin 3 (CALM3), B-cell lymphoma/leukaemia 11B (BCL11B), Cut Like Homeobox 1 (CUX1), Empty Spiracles Homeobox 1 (EMX1), Fibroblast Growth Factor Receptor 2 (FGFR2), Interleukin 6 (IL-6), Mucolipin TRP Cation Channel 1 and 2 (MCOL1/2), Polycystin 1, Transient Receptor Potential Channel Interacting (PKD1) and TNF (Tumor Necrosis Factor).

Figure 8 shows the results of this analysis as a heat map with *p*-values extracted from Pearson correlation. We noticed a positive correlation between TRPA1 and ANK1, a membrane-associated cytoskeletal protein, in GBM, KICH, KIRC, LUSC, READ and THYM tumours. BCL11B positively and significantly correlated with TRPA1 in CHOL, GBM, KICH and PRAD cancers. TRPA1 correlated with CALM3 in HSCN, KIRC, KIRP, LUAD, LUSC and PRAD. CUX1 revealed a positive correlation with TRPA1 in BRCA, HNSC, KIRC and KIRP, and a negative one in LUAD and the other homeobox gene EMX1 in BRCA, GBM, KIRC, KIRP and LUSC. FGFR2 had a positive correlation in COAD, HNSC, KIRC, KIRP and PRAD and a negative correlation in LUAD. TRPA1 and MCOL1 positively correlated in HNSC, KIRC and THYM, while MCOL2 positively correlated in CHOL, HNSC, LUSC and THYM and negatively correlated in KIRP. PAX6 positively correlated in BRCA, HNSC, KIRP, KIRC, LUSC and THYM and negatively in BLCA. PKD1 had a positive correlation with TRPA1 in HNSC, KIRP, KIRC, PRAD and THYM. The inflammatory IL-6 positively correlated with TRPA1 in BLCA, COAD, HNSC, LUAD, LUSC, PRAD, READ and THYM and negatively in KIRP. The other inflammatory factor from this analysis, TNF, correlated with TRPA1 in BLCA, BRCA, COAD, HNSC, GBM, LUAD, LUSC, PRAD and READ.

## 3. Discussion

This study offers an overview of TRPA1 gene expression and methylation and its relation to overall survival, cancer stages and metastasis, tumour purity and immune infiltration using a pan-cancer analysis of several databases. Although the description of TRP family members as temperature sensors in excitable cells for is the most accepted, in the last several decades they have been well described in non-excitable cells [17].

Members of the TRP family are expressed in many cancers, but the individual roles of TRP genes remain unclear. The TRPA1 gene consists of 27 exons and spans 55,701 base pairs of the human chromosome 8q13 [18]. The protein has six transmembrane domains (S1–S6) with the pore region between S5 and S6. From all TRP members, specific for TRPA1 is the ankyrin repeat domain that participates in protein–protein interactions and channel trafficking to the plasma membrane.

The TCGA project’s analysis, encompassing data from 31 cancer types, shows that TRPA1 is significantly overexpressed in cancers with different origins, such as kidney cancer, breast cancer, cholangiocarcinoma, head and neck squamous cell carcinoma, and lung, prostate and thyroid cancers.

Early data-driven analysis showed that TRPA1 is overexpressed in kidney tumours, and high expression indicates better patient survival [19]. In kidney renal clear cell carcinoma (KIRC), overexpression correlates with a better prognosis and disease-free survival. Although TRPA1 expression is higher in stages 2 and 3 in patients with KIRC, we calculated an increased risk using TIMER.2. Interestingly, the expression decreases in metastatic kidney cancer. From the results obtained from the evaluation of tumour infiltration, we notice that a high expression of TRPA1 in infiltrated B cells or CD8+ T cells underlies a poor prognosis. So, in clear renal cell carcinoma, the only cancer type with high expression of TRPA1 correlated with better patient survival, it is important to know the stage of the tumour and the degree of infiltration so that TRPA1 can eventually be used as a target in therapy. The increased expression would have a better outcome only in the early phase of KIRC.

A positive correlation with cancer stages was also found for BRCA and BRCA-Her2. In BRCA-Her2, TRPA1 acts as a protective factor, with its expression trending downward with increasing risk scores, as it will give a high expression in B infiltrating cells.

In lung cancer, TRPA1 expression positively correlates with higher cancer stages and metastases and indicates higher risk. These results align with a previous study showing that TRPA1 could be a novel marker for tuberculosis-related lung adenocarcinoma [20]. The analysis of tumour infiltration reveals that TRPA1 is highly expressed in the tumour environment, and higher expression in B cells improves overall survival.

TRPA1 is overexpressed in prostate adenocarcinoma without any relation to overall survival, cancer stage or metastasis. However, TRPA1 correlates negatively with tumour purity, indicating a higher expression in the microenvironment. TRPA1 was previously identified as a prostate-associated gene and acts as a modulator of angiogenesis [21], so it is more likely expressed in cells surrounding the tumoral tissue. A similar result was obtained for thyroid cancer, where TRPA1 is highly expressed only in infiltrated B cells, CD4+ T cells and dendritic cells and is correlated with better overall survival.

TRPA1 expression is lower in gastrointestinal malignancies (colon and rectum carcinoma) than in non-tumoral tissues, and it does not correlate with overall survival or disease-free survival. Other studies supported the same results and considered TRPA1 a protective predictor in these cancers [22].

Epigenetic alterations, including DNA methylation and histone modifications, modulate the expression of the TRPA1 gene, as for patients with Crohn’s disease with altered regulation of pain perception possibly related to TRPA1 promoter hypermethylation [23]. Conversely, TRPA1 may change the epigenetic profile of many genes, as shown for TRPA1 methylation in pain sensitivity [24]. The results from the UALCAN database show high levels of promoter methylation in tumoral samples compared to non-tumoral tissues in all cancer types included in this study, except GBM, KICH and KIRC.

Although the expression of the TRPA1 gene is seldom related to overall survival and disease-free survival, it is important to stress that TRPA1 is a channel permeable to Ca^2+^ only when agonists open its pore. It may also be possible that the expression relates to other signalling pathways, but we do not have samples from patients to support this assumption.

We have few pharmaceutical tools available if we consider the TRPA1 gene as a target and try to modulate its expression. Formaldehyde and dexamethasone elevate TRPA1 mRNA levels [25] and menthol may decrease them [26]. The glucocorticoid (GC) steroid dexamethasone (Dex) is used as a co-treatment to reduce the side effects of chemotherapy [27]. The same study performed in cell cultures shows that Dex decreases the proliferation of lung tumoral cells.

However, TRPA1 protein should be functional and prone to modulation to be considered a target for therapies.

A huge number of compounds selectively activate or inhibit the channel pore. Endogenously, the channel may be activated by reactive oxygen species in metastatic colorectal carcinoma cells [8] or breast and lung cancer [28], and even in tumours with a low TRPA1 expression, activating or blocking the channel may have therapeutic potential, as was proved for glioblastoma [10], osteosarcoma [7] and pancreatic cancer [5].

Allyl isothiocyanate (AITC), a dietary phytochemical compound found in some cruciferous vegetables and also a selective and potent agonist of TRPA1, was shown some years ago to decrease the viability and angiogenesis in hepatoma cells [29]. The authors did not correlate their results with TRPA1 expression. Still, they proposed an applicable therapy by using AITC-conjugated silicon quantum dots (AITC-SiQDs), which showed similar anti-cancer properties to AITC at high doses; also, in glioblastoma cell lines, AITC induced apoptosis and suppressed glioblastoma growth in a mouse model, in line with the results presented in this paper, as TRPA1 is underexpressed in glioblastoma tumours and may have protective roles [10]. In human breast cancer cells positive for oestrogen and progesterone receptors, some studies have shown that AITC decreased cell viability and stimulated apoptosis [30]. The compound inhibited colorectal tumour growth in low concentrations, as would be related to the low expression in this cancer type [31].

Differential responses were noticed for TRPA1 inhibitors. For instance, from the group of coumarins, natural compounds described previously as antitumoral compounds [32], the 3-phenyl-coumarins and resveratrol inhibited TRPA1 in a cancer cell line [33]. The specific inhibitor of TRPA1, A967079, promoted cell migration in pancreatic adenocarcinoma. Moreover, resveratrol inhibits the inflammatory cytokines Il-6 and TNF-α [34], which we found to correlate with TRPA1 regarding expression.

Studies indicating protein–protein interactions between TRPA1 and inflammatory or growth factor receptors reveal an even more complicated picture [30]. The heatmap of the Pearson correlation returned by TIMER.2 shows significant positive correlations with TNF in colon, rectal and breast cancer and glioblastoma, which implies a high stimulation of TRPA1 in inflammatory conditions. Also, IL6 positively correlates with TRPA1 in many cancers but negatively in kidney renal papillary cell carcinoma (KIRP). Both inflammatory cytokines TNF (α) and IL6 regulate bone cancer pain via TRPA1 [31], but no data have proven the relationship between TRPA1 and tumours.

AITC can inhibit the inflammatory pathway JNK/NF-κB/TNF-α, as was proved in other studies of neuroinflammation [35].

A previous study showed that the FGFR2 protein physically interacts with TRPA1 in LUAD [30], and its negative correlation (also obtained in this study for LUAD) hinders the metastatic process. These two genes positively correlate in different kidney cancer types, and in colon and prostate cancer.

The correlation with ANK1 supports the implication of TRPA1 in migration, as shown in some studies [36,37,38,39]. Elevated expression levels of PAX6, BCL11B, MCOLN2, CUX1 and EMX1 reported in colorectal cancer positively correlate with TRPA1 in other malignancies, with a possible implication in tumorigenesis [15].

## 4. Materials and Methods

### 4.1. Expression Analysis of TRPA1

We used the Diff-Exp module in the tumour immune estimation resource (TIMER) database (https://cistrome.Shinyapps.io/timer/, TIMER v2.0 | © X Shirley Liu Lab 2020 | Dana Farber Cancer Institute, Boston, MA, USA, accessed on 27 October 2024) to reveal the differential expression patterns between normal and adjacent tumour tissues for each gene of interest in all TCGA tumours. Distributions of gene expression levels are displayed using box plots, and the statistical significance of differential expression is evaluated using the Wilcoxon test; *p*-value < 0.05 is considered significant.

### 4.2. Survival Analysis of TRPA1

We used GEPIA (http://gepia.cancer-pku.cn, developed by Zefang Tang, Chenwei Li and Boxi Kang of Zhang Lab, Peking University, China, accessed on 4 September 2024) to analyse RNA expression based on data from TCGA and the Genotype-Tissue Expression project. We performed overall survival (OS) and disease-free survival (DFS) analysis for tumours with significant TRPA1 expression, as determined previously. GEPIA uses the Mantel–Cox test for the hypothesis test. In the survival analysis, each median expression of log10 (transcripts per million) was set as the cutoff to divide the patients into high- and low-expression groups. *p*-value < 0.05 was set as the cutoff criterion.

### 4.3. Correlation Analysis of TRPA1 Expression and Cancer Stages or Metastasis

We used the Gene Outcome Module from TIMER.2 (http://timer.cistrome.org/, accessed on 20 October 2024) to explore the correlation with cancer stages (S2–S4) with the flexibility to correct for multiple covariates. Once variables are submitted, TIMER performs the Cox regression. The heatmap table generated by the Gene Outcome Module shows the survival analysis of TRPA1 expression adjusted by the clinical-stage factor.

The correlation between TRPA1 expression and TNM was performed with https://tnmplot.com, A5 Genetics Ltd., accessed on 30 August 2024) which allows for online analysis of The Cancer Genome Atlas (TCGA), Genotype-Tissue Expression and Gene Expression Omnibus (GEO) data. The software generates the *p*-values of two-tailed Dunn’s tests between normal vs. tumoral and metastasis vs. tumoral tissues. We illustrate the heatmap of TRPA1 gene expression in each cancer type and the increase in median fold change (FC) values between paired samples.

### 4.4. Correlation Between TRPA1 and Immune Cells

We used the web server TIMER to study the correlation between the TRPA1 gene and immune cells (https://cistrome.shinyapps.io/timer, © X Shirley Liu Lab 2020|Dana Farber Cancer Institute, Boston, MA, USA, accessed on 20 September 2024). We determined the Spearman correlation between TRPA1 expression levels and tumour purity. TIMER generates scatterplots showing the purity-corrected partial Spearman’s rho value and statistical significance. TIMER also returned Kaplan–Meier plots for immune infiltrates and TRPA1 to visualise the survival differences. Levels are divided into low and high levels by a user-defined slider. Each plot shows a *p*-value of the log-rank test for comparing the survival curves of the two groups. The immune cells included T cells, B cells, macrophages, dendritic cells (DCs), NK cells, monocytes, mast cells, eosinophils and neutrophils. The correlation between them was shown using a heat map.

### 4.5. Hypermethylation of TRPA1 Gene Promoter with UALCAN

UALCAN (https://ualcan.path.uab.edu, The University of Alabama at Birmingham, AL, SUA, accessed on 17 July 2024) provides access to graphs and plots depicting gene expression and survival curves, evaluates promoter DNA methylation information and performs pan-cancer gene expression analysis. In this study, we also used UALCAN to analyse the promoter methylation of different genes expressed as the Beta value, which indicated their level of DNA methylation ranging from 0 (unmethylated) to 1 (fully methylated). The Beta value is the ratio of the methylated probe intensity and the sum of methylated and unmethylated probes. Different Beta value cutoffs have been considered to indicate hypermethylation (Beta value: 0.7–0.5) or hypo-methylation (Beta-value: 0.3–0.25).

### 4.6. Correlation of TRPA1 with Other Genes

We used the Gene-Corr module from TIMER.2 to explore the correlation between TRPA1 and various genes in the cancer types of interest (http://timer.cistrome.org/, accessed on 22 October 2024). The heatmap gives the purity-adjusted partial Spearman’s rho value as the degree of their correlation.

## 5. Conclusions

In summary, TRPA1 expression is modulated in many cancers and may play an essential role in the development of many malignancies, with implications for patient outcomes. However, every cancer type has a different expression and modulation of TRPA1, with the most common trait being the hypermethylation of the gene promoter. If TRPA1 is considered a target for therapies, the activation/blockade of the channel may lead to changes in migration, Ca^2+^ or inflammatory signalling pathways. Equally important is the infiltration of immune cells, which may change patient outcome. The channel function is modulated by various natural or synthetic compounds, which have already been proven in other contexts to modulate inflammatory or tumorigenic pathways; therefore, in tumours where the protein is expressed, they may be good adjuvants in cancer therapies.

Despite the extended bioinformatic analysis, this study lacks animal experiments to verify the functional assumptions. Still, this analysis may help to bring more insights into the growing evidence that TRP channels participate in carcinogenesis and may act as markers or targets in therapies.

## Figures and Tables

**Figure 1 pharmaceuticals-17-01657-f001:**
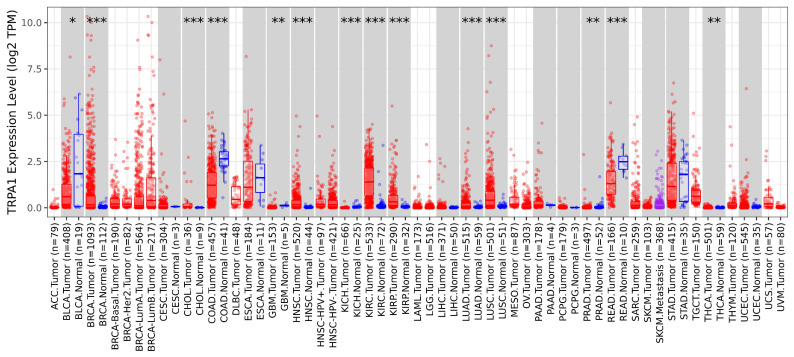
Comparisons of the TRPA1 expression in tissues from TCGA based on TIMER.2. The Y axis represents the log2 (TPM + 1) for gene expression. The blue bars indicate the non-tumoral tissues and the red bars show the tumoral tissues. TPM: transcripts per kilobase million. TRPA1 is upregulated or downregulated in the tumours compared to normal tissues for each cancer type, as displayed in grey columns when normal data are available. The statistical significance computed by the Wilcoxon test is annotated by the number of stars (*: *p*-value < 0.05; **: *p*-value < 0.01; ***: *p*-value < 0.001).

**Figure 2 pharmaceuticals-17-01657-f002:**
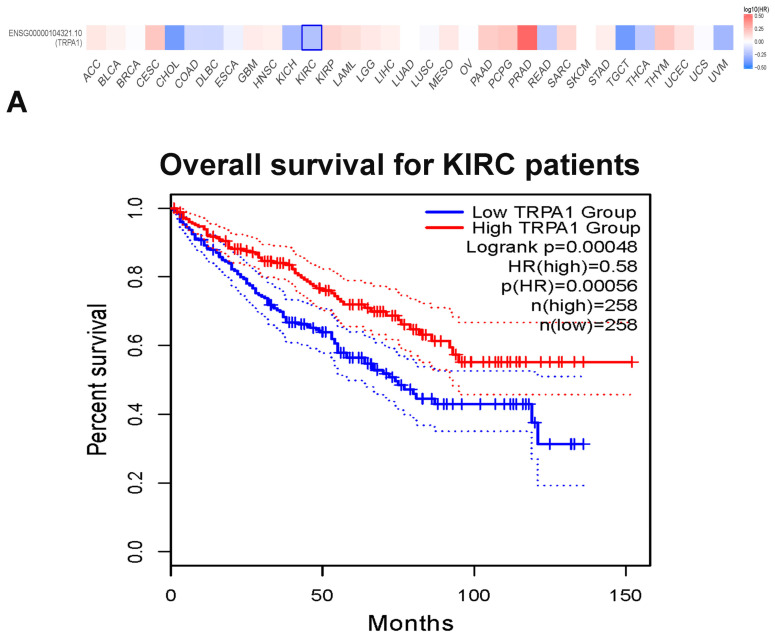
Survival significance maps and curves of TRPA1 in a pan-cancer perspective extracted from GEPIA, showing (**A**) the overall survival map and the overall survival curve of TRPA1 in KIRC patients and (**B**) the disease-free survival analysis and DFS curve of TRPA1 in KIRC and SARC patients. The results are based on the Cox proportional hazard ratio (HR). The red and blue blocks denote higher and lower risks, respectively; the rectangles with frames indicate significant unfavourable and favourable results.

**Figure 3 pharmaceuticals-17-01657-f003:**
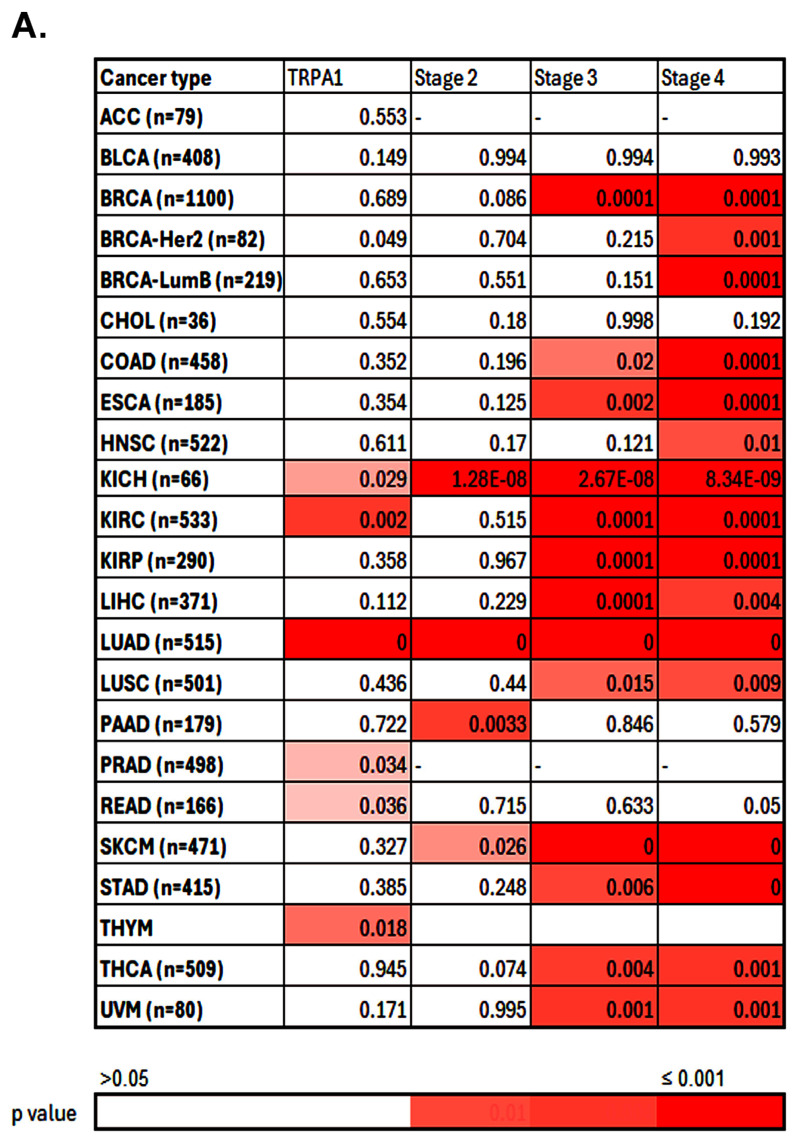
Pan-cancer analysis for cancer-stage-associated TRPA1 expression using the Timer 2.0 Exploration component. (**A**) Heatmap illustrating the correlation between TRPA1 and the cancer stages across various cancer types. (**B**) The heatmap table generated by the Gene Outcome Module shows the survival analysis of TRPA1 expression adjusted by the clinical-stage factor. The red values specify a statistically significant increased risk, whereas blue indicates a significantly decreased risk (Z-score, *p* < 0.05). The Kaplan–Meier (KM) curves provide detailed information about the corresponding significant relationship between the gene expression and survival of patients at low and high expression.

**Figure 4 pharmaceuticals-17-01657-f004:**
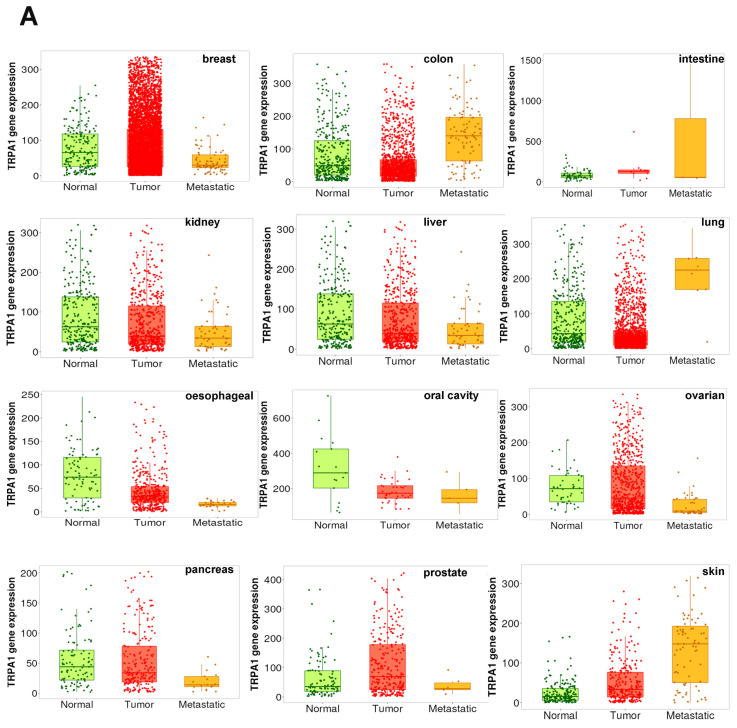
TRPA1 is differently expressed in tumour and metastatic samples. (**A**) TRPA1 gene expression in different cancers, comparing paired normal, tumour and metastatic tissues from gene chip and RNA-seq. data at TNMplot.com. (**B**) Heatmap of the two-tailed Dunn’s test probability analysing TRPA1 gene expression between normal vs. tumoral and metastasis vs. tumoral tissues in each cancer type. ** *p* < 0.01; *** *p* < 0.001; **** *p* < 0.0001. Median fold change (FC) values’ increases between paired samples are illustrated.

**Figure 5 pharmaceuticals-17-01657-f005:**
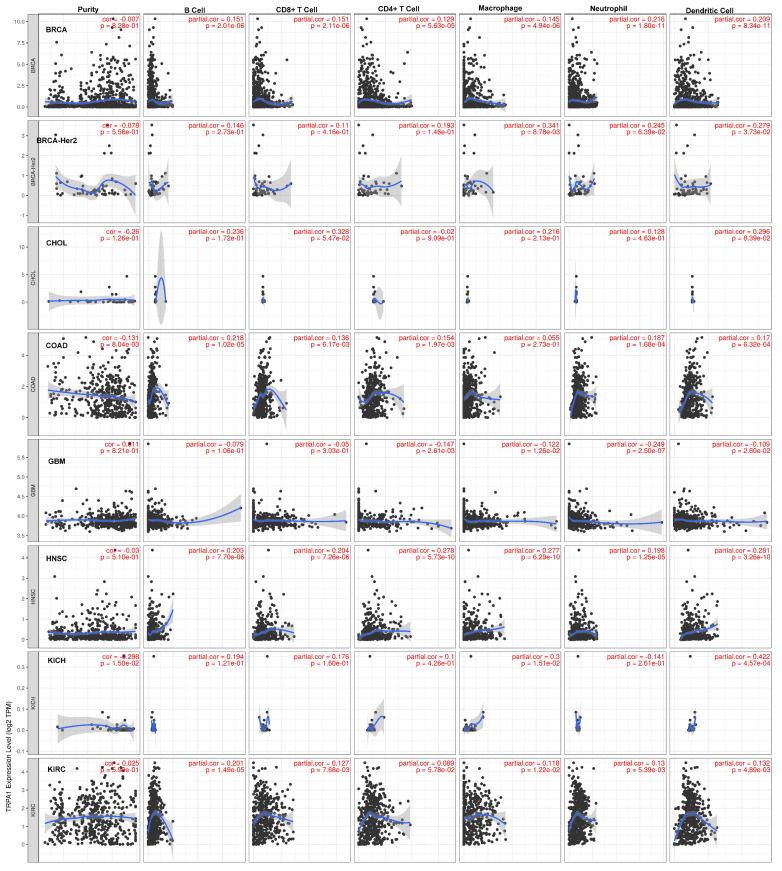
The relationship between TRPA1 expression level, tumour purity and immune cell infiltration was explored using the TIMER database. Black dots represent the number of immune cell infiltration. The blue line represents the relationship between immune cell infiltration and gene expression level. The blue line upward slope represents the positive relationship, the blue line downward sloping represents the negative relationship.

**Figure 6 pharmaceuticals-17-01657-f006:**
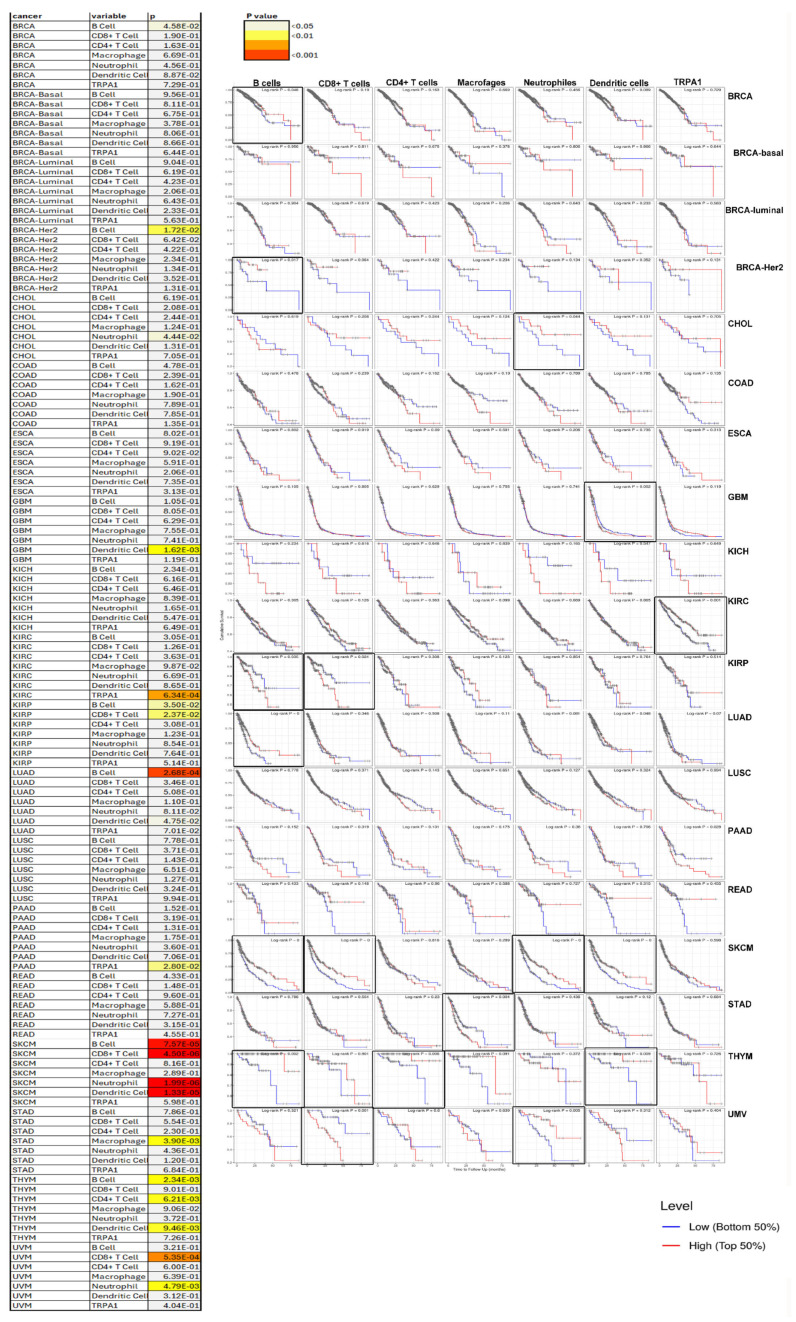
TRPA1 expression in immune infiltrates influences patients’ survival. The left side presents the *p*-value of the log-rank test for comparing the survival curves of these two groups in a heatmap. The right side displays Kaplan–Meier curves for the corresponding immune infiltrates and cancer types. The expression of the TRPA1 gene is divided into low and high levels. The hazard ratio and *p*-value for the Cox model and the log-rank *p*-value for the KM curve are shown on the KM curve plot, and the significant results are framed.

**Figure 7 pharmaceuticals-17-01657-f007:**
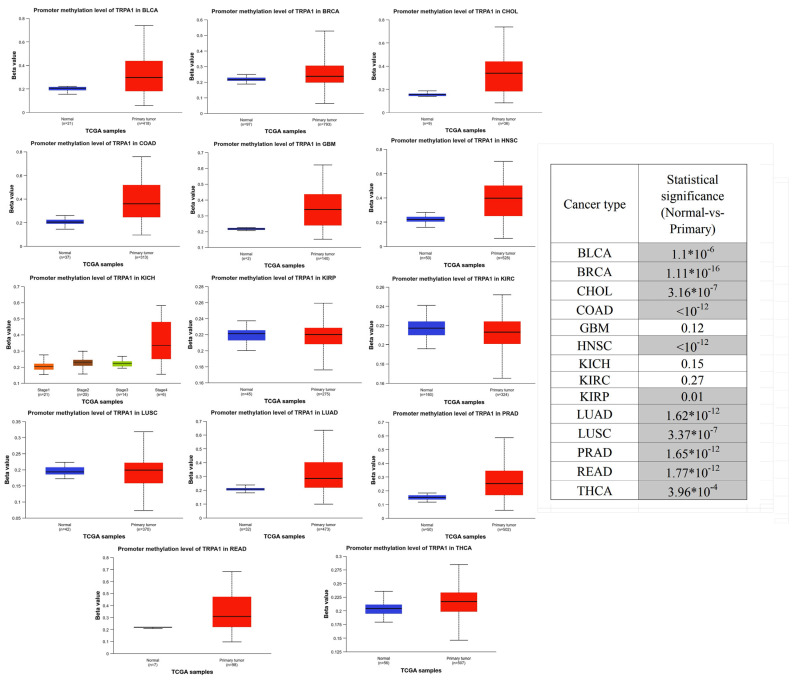
TRPA1 promoter methylation in non-tumoral (blue) and tumoral tissues (red). The plots are extracted from the UALCAN database, and *p*-values are estimated from Student’s *T* test, considering unequal variance. The significant values (*p* < 0.05) are highlighted in grey.

**Figure 8 pharmaceuticals-17-01657-f008:**
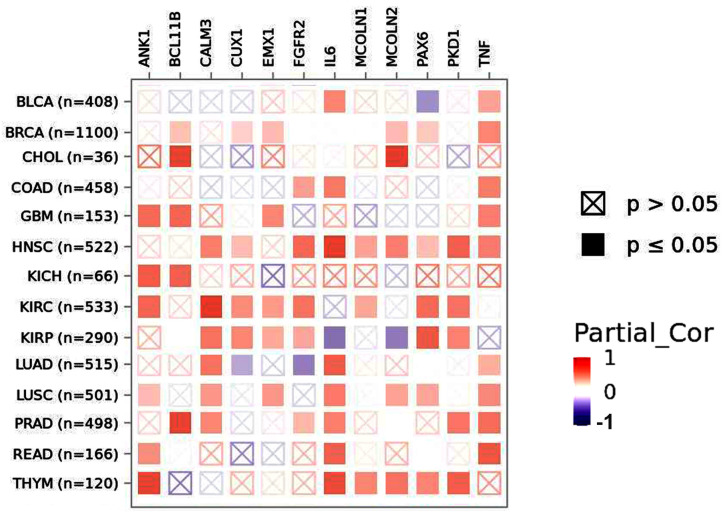
Heatmap of the Pearson correlation coefficients between TRPA1 and selected genes. Significant positive correlations are marked in red and negative ones in blue (colour scale at the right).

## Data Availability

Primary data from individual responses described herein will be made available upon request. Individual respondents will remain anonymous. All other data and materials from this study are included in this report.

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
