# Peer review of "The Potential of TRPA1 as a Therapeutic Target in Cancer—A Study Using Bioinformatic Tools"

_pharmaceuticals, 2024, doi:10.3390/ph17121657_

Round 1

Reviewer 1 Report

Comments and Suggestions for Authors • Dr Dana Cucu's review paper addresses the correlation between some clinical parameters of patients' cancer stages, metastases, and the degree of tumor infiltration and the expression of the TRPA1 channel. In particular, the main goal was to analyze a wide range of database information regarding the expression of TRPA1 in patients related to overall survival, cancer stage and metastasis, immune infiltration, and correlation with genes essential in inflammation and oxidative stress. • The topic is original and relevant to this area since it contains a huge amount of information that is statistically processed and analyzed using appropriate methods. The paper is well-written with normal/good English. • Compared to other publications, the author based on clinical material, here revealed new data on the important role of the TRPA1 channel in many malignancies and cancers and confirmed the well-known idea that the channel is one of the principal targets for therapies. The latter means activation of the TRPA1 by various natural or synthetic compounds or blockage by antagonists changing different cellular and molecular signaling pathways. For further analysis, animal experimentation data should be added to approve evidence on the role of TRPA1 in carcinogenesis. • Conclusions are consistent with the data and main arguments presented in the article. • References are appropriate.
The references should be provided at the end of the Discussion (lines 293-296), at the end of both sentences.

At the beginning of the paper, the authors should give a List of abbreviations.

This is a good article with original figures. 

Author Response

I thank the reviewer for carefully reading the paper and providing the suggestions.

Comment 1: The references should be provided at the end of the Discussion (lines 293-296), at the end of both sentences.

Response 1: I added references 36-39.

Comment 2: At the beginning of the paper, the authors should give a List of abbreviations.

Response 2: I added the abbreviation list.

Reviewer 2 Report

Comments and Suggestions for Authors

The manuscript describes an in silico investigation utilizing the TCGA database. Several analytic modules are utilized e.g. TIMER, TIMER.2 and GEPIA. Unlike other similar works in recent literature focusing on particular types of malignancies, the manuscript provides an excellent overview of the expression of TRPA1 in various types of cancer. Not only expression patterns are investigated, but potential correlation with survival and metastasis. TRPA1 expression was evaluated in tumor-infiltrating immune cells, too. Promoter methylation and potential mechanism of the effect of TRPA1 on cancer outcome are assessed. The discussion introduces relevant papers using animal models or cell cultures in a similar setting. Overall, the manuscript is well-written and highly relevant. I recommend to publish it.

Author Response

I thank the reviewer for reading the paper and appreciating my work.